palaeontology/biochemistry/molecular biology

palaeoproteomics, fossils, humic substances, protein, mass spectrometry, moa

**Author for correspondence:**
Elena R. Schroeter
e-mail: easchroe@ncsu.edu

# Proteomic method to extract, concentrate, digest and enrich peptides from fossils with coloured (humic) substances for mass spectrometry analyses

Elena R. Schroeter[1], Kevin Blackburn[2], Michael B. Goshe[2] and Mary H. Schweitzer[1]

[1]Department of Biological Sciences, and [2]Department of Molecular and Structural Biochemistry, North Carolina State University, Raleigh, NC 27513, USA

ERS, 0000-0003-4314-2976; KB, 0000-0001-8623-3902; MHS, 0000-0002-0427-3829

Humic substances are breakdown products of decaying organic matter that co-extract with proteins from fossils. These substances are difficult to separate from proteins in solution and interfere with analyses of fossil proteomes. We introduce a method combining multiple recent advances in extraction protocols to both concentrate proteins from fossil specimens with high humic content and remove humics, producing clean samples easily analysed by mass spectrometry (MS). This method includes: (i) a non-demineralizing extraction buffer that eliminates protein loss during the demineralization step in routine methods; (ii) filter-aided sample preparation (FASP) of peptides, which concentrates and digests extracts in one filter, allowing the separation of large humics after digestion; (iii) centrifugal stage tipping, which further clarifies and concentrates samples in a uniform process performed simultaneously on multiple samples. We apply this method to a moa fossil (approx. 800–1000 years) dark with humic content, generating colourless samples and enabling the detection of more proteins with greater sequence coverage than previous MS analyses on this same specimen. This workflow allows analyses of low-abundance proteins in fossils containing humics and thus may widen the range of extinct organisms and regions of their proteomes we can explore with MS.

# 1. Background

Humic substances are breakdown products of decaying organic matter [1] that, to the frustration of many protein researchers, co-extract with proteins from fossils (e.g. [2]) and soil (e.g. [3]). These complex, hydrophilic, dark-coloured substances have a large molecular weight range (less than 10 kDa to greater than 100 kDa) and are notoriously difficult to separate from proteins in solution [1,4–7]. This causes a host of problems for a number of downstream chemical analyses when conducting palaeoproteomics of fossil tissues. For example, these substances interfere with colorimetric assays used for protein quantitation (e.g. Bradford and bicinchoninic acid (BCA) assays) [8], preventing accurate measurement of protein for subsequent analyses. They also bind silver-stain, generating false-positives in sodium dodecyl sulfate–polyacrylamide gel electrophoresis (SDS-PAGE) gels, or can pre-stain gel lanes and obscure chemical stains applied to separated proteins [9,10]. They can contaminate the isotopic content of ancient bone and interfere with stable isotope analyses [2,11]. In mass spectrometry (MS) experiments, humic substances can cause ion suppression during electrospray ionization (ESI) [3,12], leading to either depressed signal intensity or no signal (regardless of the presence of protein in the sample). Further, they can clog analytical columns or spray tips during liquid chromatography [3], causing unstable spray and/or the loss of samples, columns or spray tips. Even when these substances do not clog analytical columns, they can generate a build-up of particulate matter on the ion optics, which can suppress or prevent signal during analyses of all subsequent samples, necessitating expensive and time-consuming instrument cleaning.

Although the analytical problems generated by humic substances can be quite severe, fossilized bone containing high amounts of humic substances may potentially be valuable sources of palaeoproteomic information. Humic acids can play a role in diagenetically cross-linking collagen molecules in a manner similar to leather tanning or fixation with formaldehyde [1]. This suggests that fossils with humic content may be *more* likely to preserve original, minimally altered collagen molecules incorporated into diagenetic complexes with exogenous humics [1]. However, any such preserved proteins are not readily accessible by MS, requiring them to be both extracted and separated from interfering humic substances.

A number of approaches have been employed to remove humic substances from bone [2,13] and soil [3] samples. For example, Szpak *et al.* [2] tested the efficiency of both sodium hydroxide (NaOH) and ultrafiltration to remove humics from archaeological bone for isotopic analyses and found that post-demineralization with HCl, sequential incubations with NaOH could be used to remove the alkali-soluble fraction of humic substances from samples. Qian & Hettich [3] developed a method that, subsequent to enzymatic digestion of soil protein extracts, used ultrafilters to remove humics that had been precipitated from an acidified (approx. pH 3) peptide digest. Cleland [13] employed a method using both a non-demineralizing buffer and magnetic beads to separate humics from proteins in unconcentrated protein extracts.

Here, we introduce a method that builds on these findings and several recent advances in protein extraction protocols to optimize the recovery of ancient proteins for palaeoproteomics from fossils that contain a high amount of humic substances. This method (i) uses a non-demineralization extraction buffer [14], because the demineralization phase of traditional bone protein extractions has been shown to be a source of loss [15], particularly for non-collagenous proteins (NCPs); (ii) allows for the concentration of a relatively large amount of extraction supernatant into a small sample volume, which is crucial for fossil specimens with a low abundance of preserved protein, but challenging for those that also contain humic substances, which co-elute with protein; (iii) uses a modified filter-aided sample preparation (FASP) protocol [16] to filter humic substances from proteins/peptides both before and after digestion; (iv) removes remaining humic substances with centrifugal stage-tip protocol [17] that can prepare multiple samples simultaneously and consistently. Although the individual components of this method have been variously applied in previous palaeoproteomic studies (e.g. non-demineralizing buffer [13,18], FASP [19,20]), their combination and adaptation for humic removal has not yet been explored. We apply this combined method to a moa fossil [21] with humic content, from which preserved proteins were previously characterized by liquid chromatography–tandem mass spectrometry (LC–MS/MS) [22], to demonstrate its utility for preparing fossils samples for palaeoproteomic analyses.

# 2. Material and methods

## 2.1. Specimen

We analysed fossilized (800–1000 years) moa cortical bone (MOR OFT255, courtesy J. Horner and Museum of the Rockies) recovered from New Zealand cave deposits [21]. This specimen was used

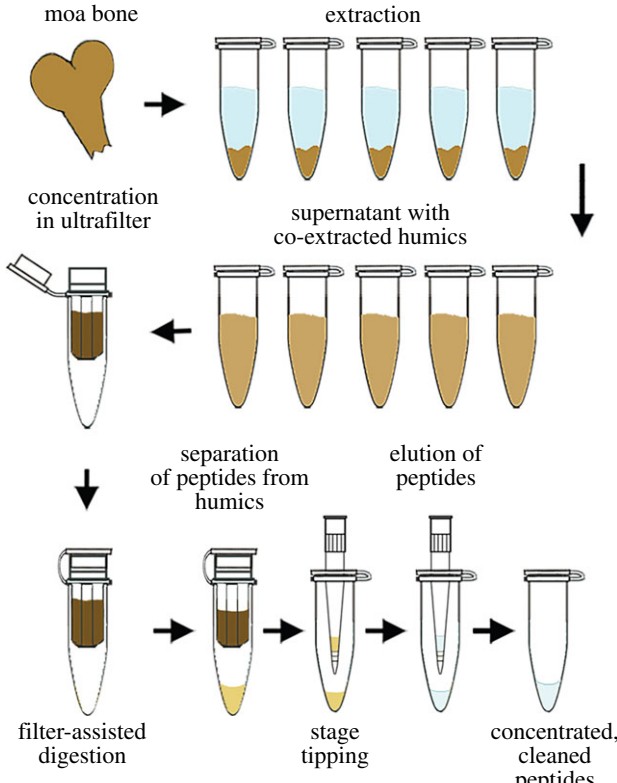

**Figure 1.** Diagram illustrating the extraction workflow of Method 2 (400/200/4 only). NaOH pretreatment step of Method 1 is not illustrated; however, NaOH supernatants were also concentrated, digested and stage tipped as shown.

because (i) despite its relatively young age, the dark coloration of both the whole cortical fragments (described as very dark brown to nearly black) [23] and the extraction supernatants produced in previous studies [22,23] suggests a high content of humic substances; (ii) previous studies of this specimen using different extraction methods [23] and MS analyses [22] successfully produced evidence of protein preservation, including collagen I alpha 1 and alpha 2 sequences with biological and diagenetic post-translational modifications (PTMs). Thus, MOR OFT255 represents an ideal test case, in which both bone proteins and humic substances are known to be present in the tissue, allowing us to monitor the effect of humic removal on peptide recovery.

## 2.2. Protein extraction

Two extraction methods were tested: one that used a single incubation in a non-demineralization reagent [14] for protein solubilization, and one that incorporated a brief pretreatment of the bone powder with NaOH, followed by an extraction identical to the first in all other respects. NaOH has been used to remove 'pigments' from bone prior to collagen extraction [24], humic acids from archaeological bone [2] or to improve the efficiency of some bone protein extraction methods [15]. Therefore, we tested this extraction method both with and without an NaOH pretreatment step to assess its usefulness in both removing humics and extracting peptides. Figure 1 illustrates the basic workflow of the described extraction procedure (NaOH pretreatment step not shown). Electronic supplementary material, figure S1 illustrates a diagram of supernatant collection steps and sample generation for each method. Electronic supplementary material, table S1 provides a concise breakdown of the steps comprising the extraction, FASP and stage-tipping procedures.

A piece of dark cortical bone was ground to a fine powder in a sterilized mortar and pestle. Bone powder (50–55 mg tube$^{-1}$) was aliquoted into 10 1.5 ml Protein LoBind centrifugation tubes (Eppendorf). An equal number of empty tubes, containing no bone powder but treated identically to bone samples in every way, were also prepared (i.e. 'blank' control samples). Five tubes of each set received 1 ml of 0.1 M NaOH (pH ~12.8), were vortexed to mix, then were incubated at 4℃ for 4 h with rocking. The remaining five aliquots and blanks were stored dry at 4℃ for the same length of

time. Samples were then centrifuged at 15 000 r.c.f. (relative centrifugal force) for 15 min, and NaOH fractions were collected and frozen at −80°C to await concentration.

All 10 aliquots of bone powder then received 600 µl of a solution [14] containing 400 mM ammonium phosphate diabasic (APD), 200 mM ammonium bicarbonate (ABC) and 4 M guanidine HCl (GuHCl) (pH 8.2), hereafter referred to as '400/200/4' for brevity. Samples were vortexed to mix, incubated at 75°C in a heat block overnight, then subsequently centrifuged at 15 000 r.c.f. for 15 min and the 400/200/4 supernatant was collected. Fractions of 400/200/4 that followed NaOH pretreatment were kept separate from those that had been incubated with an untreated bone to enable comparison. Supernatants were frozen at −80°C to await concentration.

## 2.3. Extract concentration and digestion

Supernatants were pooled into three distinct samples, each representative of the extract from approximately 250 mg of bone powder (or five blank tubes): (i) 0.1 M NaOH (approx. 5 ml of supernatant), (ii) 400/200/4 that had been applied after the NaOH pretreatment (approx. 3 ml, hereafter 'NaOH-400/200/4'), and (iii) 400/200/4 from untreated bone (approx. 3 ml). The NaOH and NaOH-400/200/4 samples, together, comprise 'Method 1', and the 400/200/4 sample from untreated bone represents the sole incubation of 'Method 2'. All samples were concentrated and buffer exchanged at room temperature using 3 kDa molecular weight cut-off (MWCO) centrifugal filters (Amicon Ultra; Millipore). For each sample, approximately 400 µl of supernatant was added to a filter, then centrifuged for 20 min at 14 000 r.c.f. The flow-through was then discarded and the process repeated in increments of 300 µl (to accommodate sample held back in the filter without overfilling to the lip of the seal) until all supernatant (3–5 ml) had been concentrated into a single filter unit. Only one filter was used per sample to reduce the adsorptive loss of protein on the filter membranes. After the last centrifugation, filters containing concentrated sample received 300 µl of 50 mM ABC (pH 7.8) and were then stored overnight at 4°C, in a humidity chamber wrapped in parafilm to prevent filters from drying. Samples were centrifuged at 14 000 r.c.f. for 30 min at room temperature, then buffer exchanged into 300 µl ABC twice more, mixing the ABC in the solvent reservoir of the filtration device prior to centrifugation by pumping the solution in and out of the pipette tip several times to ensure the whole sample had been mixed.

Concentrated samples were prepared for LC–MS/MS analysis using a modified version of the standard FASP protocol [16]. Each sample received 50 µl of 10 mM DTT, was incubated in a heating oven at 60°C, then centrifuged at 14 000 r.c.f. for 15 min, followed by two exchanges of 200 µl of 8 M urea (in 100 mM Tris, pH 8.5) and centrifugation for 15 min at 14 000 r.c.f. Samples were alkylated with 100 µl of 50 mM iodoacetamide (in 8 M urea) for 20 min in the dark at room temperature, then incubated with three exchanges of 200 µl of 8 M urea followed by centrifugation for 20 min at 14 000 r.c.f. Samples were then exchanged three times into 300 µl of 50 mM ABC followed by centrifugation for 30 min at 14 000 r.c.f.

Filters were transferred into new collection tubes, and each sample received 500 ng of trypsin solubilized in 50 µl of 50 mM ABC (well-mixed with the sample in the filter by pumping the pipette as described above). Samples were allowed to digest overnight in a humidity chamber at 37°C in a heating oven. After digestion, filters were centrifuged at 14 000 r.c.f. for 30 min. Then, 40 µl of 50 mM ABC was added to the filter, mixed with the pipette to suspend any sediment at the bottom, and samples were centrifuged again at 14 000 r.c.f. This step was then repeated twice, and the final filtered sample of approximately 150 µl was transferred to a new 1.5 ml Protein LoBind centrifugation tube and received 3 µl of formic acid (FA) to quench any further digestion and precipitate any remaining acid-insoluble humics out of solution while allowing peptides to remain suspended [3]. Samples were then frozen at −80°C to await stage tipping.

## 2.4. Sample purification and concentration

Samples were stage tipped using self-made tips following a protocol by Rappssilber et al. [25] and an adapted centrifugal protocol proposed by Yu et al. [17] Briefly, stage tips were assembled by placing two discs of C18 membrane (Empore, 3M), perforated from a larger sheet with a blunt-point, 16 gauge needle (I.D. 1.19 mm, Hamilton), in the tip of a 200 µl pipette tip (electronic supplementary material, figure S2A). Stage tips were then placed in a collection tube made by perforating the top of a 1.5 ml Protein LoBind tube with a heated glass Pasteur pipette (Fisher) (electronic supplementary material, figure S2B). By placing stage tips in collection tubes (electronic supplementary material, figure S2C), multiple samples were stage tipped simultaneously and uniformly using a centrifuge. Three stage-tip

assemblies received 20 µl of methanol, and were then centrifuged for 30 s at 1000 r.c.f. Tips were wetted with 20 µl of 80% acetonitrile (ACN)/0.2% FA followed by centrifugation for 30 s at 1000 r.c.f., then equilibrated with 20 µl of 0.2% FA and centrifuged for 1 min at 1000 r.c.f. Tips were placed in new collection tubes, and each concentrated sample (approx. 150 µl each) was then passed through a stage tip by centrifugation at 1000 r.c.f. for 3 min. The filtered sample from the collection tube was removed and passed through the same stage tip to allow an additional opportunity for peptides to bind to the C18 membrane. After this second pass, the flow-through (2× filtered sample) was collected and stored at 80°C. Stage tips were then washed with 40 µl of 0.2% FA, centrifuged for 2 min at 1000 r.c.f., then the tips were moved to new collection tubes and peptides were eluted with 20 µl of 80% ACN/0.2% FA. The peptide elution step was repeated, resulting in a final sample volume of 40 µl. The final sample was dried under vacuum centrifugation and then stored at −80°C until LC–MS/MS analysis could be performed.

## 2.5. Liquid chromatography-tandem mass spectrometry analysis

Samples were analysed by LC–MS/MS analysis using an Easy-nLC 1000 UHPLC (Thermo Scientific) coupled to an Orbitrap Elite mass spectrometer (Thermo Scientific). Samples were reconstituted in 25 µl of 0.1% FA, then 5 µl injections were loaded onto a new nanoViper nano trap column (Thermo Scientific; I.D. 100 µm, 2 cm, PepMap C18, 5 µm particles) and in-line separated with a new PicoFrit capillary column (New Objective; I.D. 75 µm × 25 cm) with an integrated PicoTip emitter, packed in-house with Reprosil C18-AQ (3 µm) stationary phase using a flow rate of 300 nl min$^{-1}$. For separation, the following linear gradients were applied, where mobile phase A is 0.1% FA and mobile phase B is 100% ACN/0.1% FA: $t = 0$ min, 0% B; $t = 1$ min, 7% B; $t = 120$ min, 40% B; $t = 121$ min, 95% B, $t = 136$ min, 95% B. The MS acquisition was performed in positive ion mode with the following parameters: the 10 most intense precursor ions were fragmented using high-energy collisional disassociation (HCD) (35% normalized collision energy); isolation window was 3 $m/z$; precursor and product ion resolution was set to 60 000 and 15 000, respectively; precursor scan range was $m/z$ 400–2000; both precursor and product ions were analysed in the orbitrap. Dynamic exclusion was enabled, with a repeat count of 2, a repeat duration of 30 s and an exclusion duration of 120 s. Three injections of each sample were performed, and identification results across these injections were combined for each sample.

## 2.6. Data analyses

Raw data files [26] from the LC–MS/MS acquisitions were processed using Proteome Discoverer 1.4.1.14 (Thermo Scientific) and searched against the NCBI Aves protein database (retrieved May 19, 2015) using Mascot (Matrix Science, version 2.5) and X!Tandem (The GPM, version CYCLONE 2010.12.01.1) using the following parameters: precursor mass tolerance = 10 ppm, fragment ion tolerance = 0.02 Da; missed cleavages = 2; non-specific cleavage = one end; fixed PTMs = carbamidomethylation (C); variable PTMs = oxidation (P), oxidation (K), deamidation (NQ). X!Tandem searched the additional variable PTMs: Glu > pyro-Glu @N; Gln > pyro-Glu @N; ammonia-loss @N. Protein/peptide identifications were validated in Scaffold (Proteome Software, version Scaffold_4.8.6). Peptide identifications established at greater than 95% probability [27] were accepted; protein identifications established at greater than 99% probability [28] and containing at least two unique peptides were accepted.

# 3. Results and discussion

## 3.1. Sample clarity

Although the initial moa bone extract, particularly the 400/200/4 supernatants, were dark in colour even before concentration, the final MS-ready samples were nearly colourless (figure 2) and virtually indistinguishable from blank controls that were processed using (initially) empty tubes as samples (electronic supplementary material, figure S3). Initial humic-containing supernatants (figure 2, Row 1) became increasingly dark and opaque during concentration, and the flow-through during filtering was markedly lighter than the portion of the supernatant held back in the filter (figure 3). Additionally, although 3 kDa MWCO, 500 µl Amicon Ultra filter units can concentrate 500 µl samples into an approximately 30 µl volume (and did so in the negative controls), the in-filter volume of

| -------- method 1  -------- | | method 2 |

| NaOH | NaOH-400/200/4 | 400/200/4 |

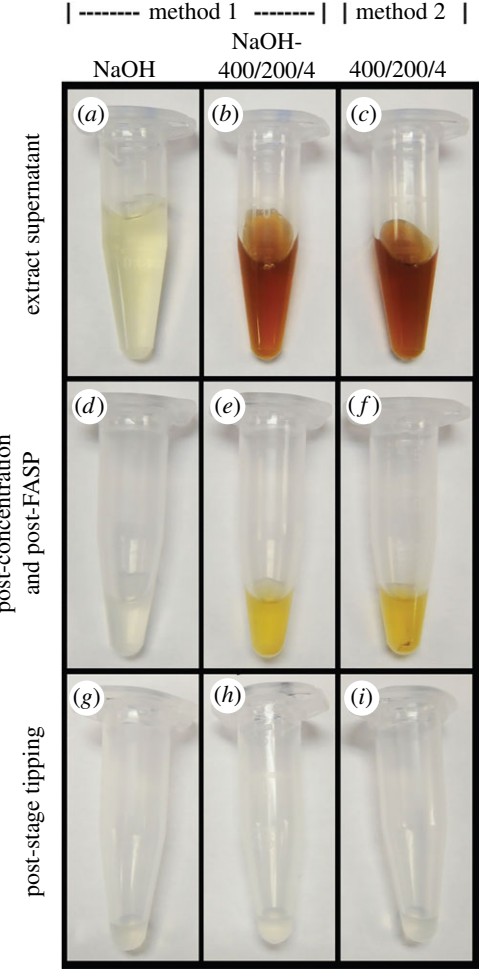

**Figure 2.** Progression of extracts through preparation for MS. (Row 1) Supernatant (NaOH or 400/200/4) immediately upon collection. Each tube represents one of five that were subsequently concentrated together. (Row 2) Supernatants after concentration and FASP using 3 kDa MWCO filters. Despite combining five tubes with high humic content, passing samples through the filter after digestion removed a large portion of the high-weight humics from the smaller peptides. (Row 3) Supernatants after stage tipping. Upon elution from the stage tips, samples were nearly colourless, suggesting that most inferring humic substances have been removed.

concentrated 400/200/4 bone samples was approximately 150 µl (figure 3). These observations indicate that the humic substances that co-extracted with the protein from the bone largely remained in the filter instead of passing through, which is consistent with the reported size of these substances (e.g. mostly greater than 10 kDa[3]) and the findings of Szpack et al. [2] that pre-digestion ultrafiltration is not useful for the separation of humics from protein extract. These humic substances remained in the filter even after digestion using the FASP protocol, allowing tryptically digested peptides (now less than 3 kDa) to be separated from these substances when passed through the filter during centrifugation. This separation was indicated by the drastic reduction in colour and opaqueness between the concentrated samples before (figure 3) and after (e.g. figure 2e,f) digestion.

Post-FASP (and pre-stage-tip) concentrated samples, while lighter in colour than the initial extracts, still possessed some coloration, suggesting that some humic and/or other substances that might cause interference in MS were still present. Subsequent stage tipping of the samples removed these last vestiges of colour, and the final eluted samples were nearly colourless (figure 2, Row 3).

Notably, although NaOH pretreatment has been reported as an effective method of removing humic substances from fossil bone [2], the single NaOH pretreatment incubation employed here was not effective at removing a substantial amount of humics from the moa bone prior to protein solubilization, as evidenced by the pale colour of this fraction (figure 2a), and the lack of difference in colour between pretreated and non-pretreated 400/200/4 fractions (figure 2b,c). This difference may be because Szpack et al. [2] applied the NaOH treatment to demineralized bone, whereas we applied it directly to ground bone powder.

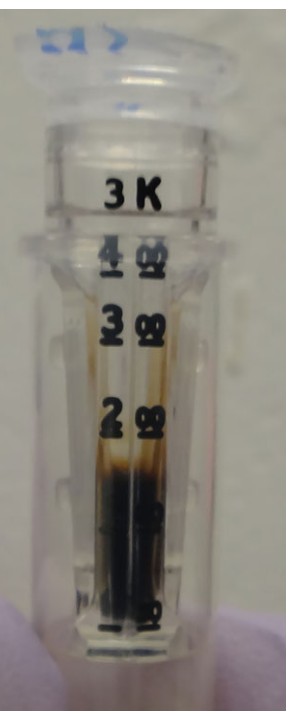

**Figure 3.** Concentrated 400/200/4 extract, before FASP digestion. Extracts dark with humic substances became increasingly dark and opaque during concentration, indicating that humic substances were too large to pass through the filter and co-concentrated with proteins. After proteolytic digestion, these humics remained in the filter while digested peptides passed through.

## 3.2. Mass spectrometry

Across all injections of moa extractions analysed by LC–MS/MS, 20 bone proteins were identified, including eight collagen alpha chains (six different collagen types) and 12 NCPs (table 1). The total number of unique peptides, total peptide spectral matches (PSMs) and sequence coverage for each fraction and method by protein are listed in electronic supplementary material, table S2. Only keratins (and no bone proteins) were identified in any of the blank (reagent) control extract injections.

Nearly all the proteins identified were recovered from both methods, which shared 18 of the 20 proteins (table 1). The exceptions were periostin, which was only found in the 'NaOH-400/200/4' fraction of Method 1, and prolargin, which was found in the '400/200/4 only' extract, or Method 2. However, because these proteins were both identified by a relatively low number of PSMs from their respective fractions (3–4 PSMs; table 1), this is probably a stochastic variation caused by data-dependent acquisition [29] and not a true difference between the methods. In fact, when the number of unique (non-overlapping) peptides and per cent sequence coverage identified from each protein is considered, the results from Methods 1 and 2 are nearly identical, with only slight variations that do not consistently favour either method (figure 4). This suggests that both methods recover roughly the same subset of the moa bone proteome.

When the two fractions of Method 1 are considered separately, the 'NaOH-400/200/4' fraction contains all the proteomic diversity obtained by this method; analysis of the 'NaOH' pretreatment extract did not identify any unique proteins compared to the subsequent fraction (table 1). Although the 'NaOH' fraction possessed a greater number of collagen I PSMs than the NaOH-400/200/4 fraction (table 1), the number of unique peptides and per cent sequence coverage for all proteins were either substantially greater in the NaOH-400/200/4 fraction, or roughly equal between the two (electronic supplementary material, table S2). This suggests that unlike demineralization treatments [15], the NaOH pretreatment is not extracting a different subset of proteomic information than the following solubilization step. Further, the nearly identical results obtained from the pretreated (NaOH-400/200/4) and non-treated (400/200/4) fractions (table 1 and electronic supplementary material, table S2) indicate that NaOH pretreatment under these conditions does not improve the capacity of the 400/200/4 to solubilize proteins from the bone matrix.

**Table 1.** Proteins identified in each fraction by LC–MS/MS analysis. Twenty proteins were identified across all injections of all fractions. Listed numbers indicate the total number of peptide spectral matches (PSMs) that were identified from each fraction per protein. (Note: n.d. indicates protein was not detected in this fraction using the LC–MS/MS analysis described in this study.)

| | Method 1 | | Method 2 |
| --- | --- | --- | --- |
| protein | NaOH | NaOH-400/200/4 | 400/200/4 |
| collagen I, alpha 1 | 763 | 643 | 702 |
| collagen I, alpha 2 | 590 | 498 | 570 |
| collagen III, alpha 1 | 18 | 24 | 28 |
| collagen V, alpha 1 | n.d. | 26 | 21 |
| collagen V, alpha 2 | 7 | 32 | 42 |
| collagen VI, alpha 3 | n.d. | 18 | 22 |
| collagen XI, alpha 1 | n.d. | 58 | 46 |
| collagen XII, alpha 1 | 4 | 23 | 26 |
| apolipoprotein A-1 | 6 | 18 | 14 |
| decorin | n.d. | 13 | 21 |
| mimecan | 16 | 35 | 38 |
| moesin | n.d. | 18 | 18 |
| osteomodulin | n.d. | 28 | 26 |
| PEDF | n.d. | 10 | 7 |
| periostin | n.d. | 4 | n.d. |
| prolargin | n.d. | n.d. | 3 |
| serum albumin | n.d. | 9 | 2 |
| sushi repeat SRPX | n.d. | 20 | 20 |
| thrombospondin-1 | n.d. | 41 | 36 |
| vitronectin | 38 | 65 | 63 |

## 3.3. Efficiency and potential variations of the experimental workflow

Previous LC–MS/MS analyses of this fossil produced sequences of collagen I, collagen II and collagen V [22]. The current method resulted in 15 additional proteins, including 12 NCPs, which ranged from 1.3 to 18.4% sequence coverage for a given protein (electronic supplementary material, table S2; figure 4b). This represents a substantial improvement over the previous study, which used a more common HCl demineralization followed by ABC solubilization, and used an entire gram of fossil tissue as opposed to 250 mg [22]. A direct comparison of the HCl-ABC method employed by Cleland *et al.* [22] and this combined method is needed to quantify the relative contributions to the observed increase in efficiency from the various differences in their workflows (e.g. extraction buffers, MS instrument, column packing) and to fully account for possible intra-bone variation in protein preservation in this specimen [30,31]. Regardless, this combined workflow obtained 15 additional proteins previously unreported for this specimen, supporting its overall improvement in recovery.

The design of this protocol as described here could potentially be modified to suit the needs of individual studies. For example, we used conical 3 kDa MWCO ultrafilters (Millipore), to ensure that small bone proteins or protein fragments were not potentially being lost. However, researchers that use 10 kDa MWCO filters for their studies (e.g. [19,20]) might consider using flat-bottomed filters, as there is some evidence that flat-bottomed ultrafilters may produce better results by eliminating the dead space in the bottom of the filter [32]. Additionally, larger centrifugal filters (e.g. 2 or 4 ml) may be used to reduce the concentration time, though the depth of the larger filters may be more difficult for performing FASP.

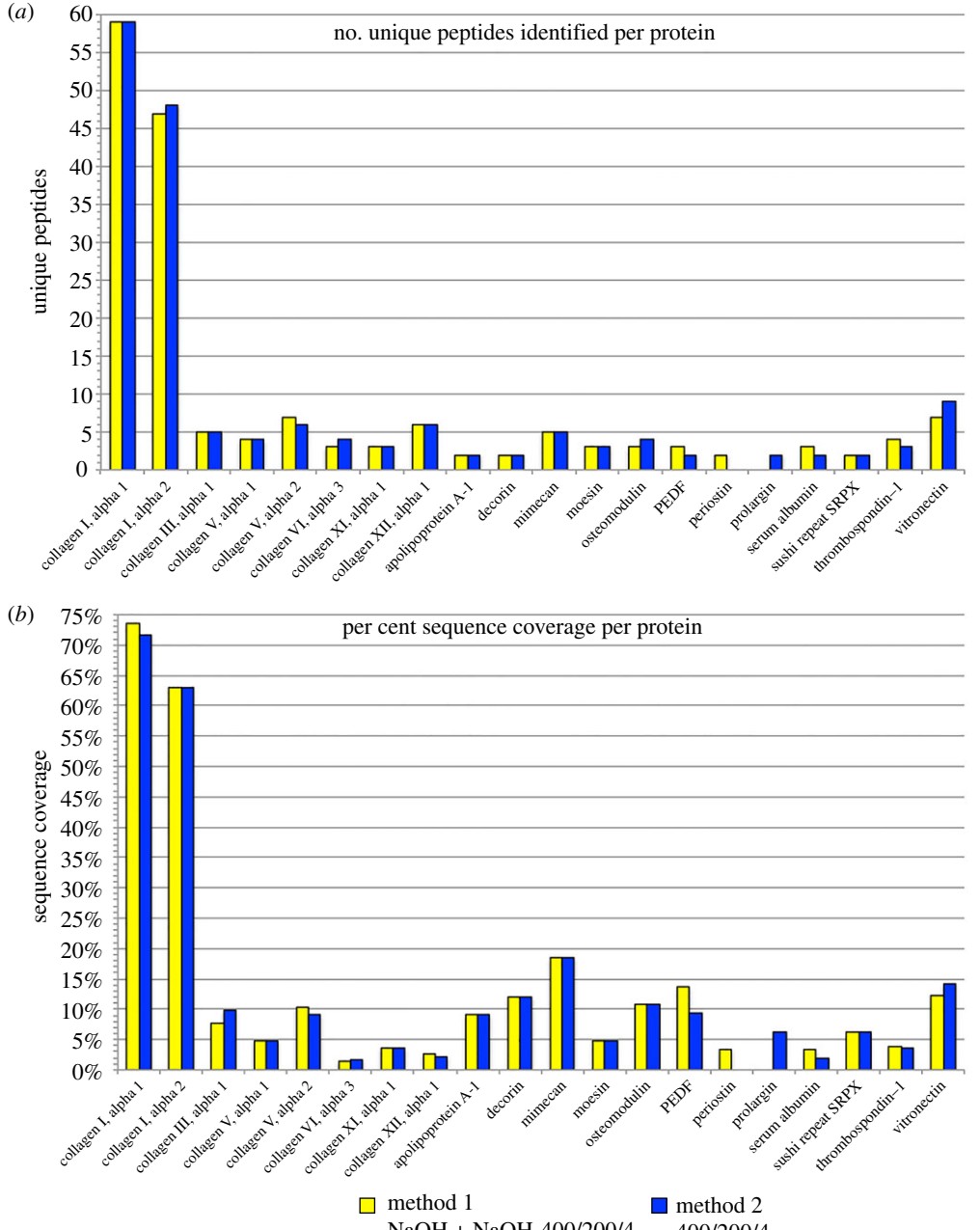

**Figure 4.** Comparison of the number of unique peptides (*a*) and sequence coverage (*b*) identified for each protein between Method 1 (NaOH + NaOH-400/200/4 fractions) and Method 2 (400/200/4 only). Both methods recover nearly identical amounts of unique peptides and sequence coverage for most proteins, suggesting that the inclusion of an NaOH pretreatment step in Method 1 does not provide additional proteome information over the sole 400/200/4 incubation of Method 2. These data represent the total, non-overlapping peptides and coverage relative to the mature form of the protein (i.e. without pro-peptide regions), combined from all three injections of each sample. See electronic supplementary material, table S2 for more details.

## 4. Conclusion

The method presented here combined a non-demineralization extraction buffer [14], filter-aided concentration and digestion of extracts and a centrifugal stage-tip protocol [17] to successfully concentrate protein extracts from moa fossil inferred from its dark colour to have a high humic content, then remove those humics to allow LC–MS/MS analysis without mechanical or ionic interference from these substances. These analyses produced peptides from 20 bone proteins, 15 of which are newly identified in this specimen versus previous analyses using a more standard approach [22]. Because this workflow allows MS analyses of low-abundance proteins in fossils with abundant

humic content, it has the potential to widen the range of extinct organisms amenable to MS analyses and increase regions of the proteome we can explore.

Although NaOH has been shown to aid in the removal of humic substances from a fossil bone pellet that has been demineralized with HCl prior to NaOH treatment [2], we found that a version of this method that incorporates a NaOH pretreatment step to the protocol (Method 1) does not remove more humics, recover more proteins or recover a substantially greater portion of the sequence of any protein than the method using only 400/200/4 (Method 2). This indicates that NaOH pretreatment is not effective for humic removal in methods that use non-demineralizing [14] solubilization reagents to avoid loss of NCPs in the demineralization fraction [15]. We therefore recommend Method 2, as it produces nearly identical results with less cost in terms of time, supplies and instrument fees.

Data accessibility. All raw mass spectrometry data and Scaffold results files are available at Dryad: http://dx.doi.org/10.5061/dryad.3tv1523 [26].

Authors' contributions. M.H.S. obtained the specimen, E.R.S. performed molecular laboratory work and prepared samples, E.R.S., K.B. and M.B.G. performed MS analyses, E.R.S. analysed and interpreted data and drafted the manuscript. All authors contributed to experimental design and gave final approval for publication.

Competing interests. The authors declare they have no competing interests.

Funding. This work was funded by an Arnold O. Beckman Postdoctoral Fellowship to E.R.S., an NSF MRI grant no. (DBI-1126244) to M.B.G. and K.B., and an NSF INSPIRE grant (EAR-1344198) to M.B.G. and M.H.S.

Acknowledgements. We thank W. Zheng for logistical support and T. Cleland and four anonymous reviewers for helpful comments on earlier versions of this manuscript.

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
