## [Reviewer comments · Royal Society Open Science]

Review History

RSOS-181433.R0 (Original submission)

Review form: Reviewer 1

Is the manuscript scientifically sound in its present form?

Yes

Are the interpretations and conclusions justified by the results?

Yes

Is the language acceptable?

Yes

Is it clear how to access all supporting data?

Yes

Do you have any ethical concerns with this paper?

No

Have you any concerns about statistical analyses in this paper?

No

Recommendation?

Major revision is needed (please make suggestions in comments)

Comments to the Author(s)

Schroeter et al. build in the current manuscript on their earlier works in an effort to mitigate a potentially major issue in palaeoproteomics - the presence of large amounts of humics in fossil bone interfering during protein extraction and MS analysis. This is a relevant question to resolve, as is indicated by the recent literature on this topic as well.

The major issue with the manuscript is that an internal control is lacking. That is, claims of relevance made in the paper of enhanced protein extraction through either Method 1 and Method 2 refer to extractions and data presented in a previous paper (Cleland et al., 2016, PRSB; Cleland et al., 2012, Plos One). However, recent research in protein preservation has shown remarkable variation of within-bone protein preservation, necessitating that extraction comparisons are performed on homogenized bone powders. Further issues with comparing the enhanced results derive from an absence in protein or peptide quantification, differences in MS set-up, and column chemistry. Therefore, the main conclusion of the paper could be seen as suggestive.

Minor comments:

-Lines 54-65. This is one massive sentence. Please break up in smaller sentences.

-Lines 85-98. The four elements of the used extraction methods are presented as novel. However, each of these has already appeared in the palaeoproteomic literature at least once. To avoid the suggestion that all four elements are introduced here for the first time, references to relevant literature are justified.

-Line 96: The re-occurring claim that this bone specimen has a particularly high humic content requires quantification.

-Line 126 vs line 147: You mention "a small piece", but later refer to approx. 250 mg., which is a large piece of bone, especially of this age. These two statements are in conflict, especially as "small" frequently refers to samples under 20mg in the isotope, collagen, and bone proteome literature.

-Line 257: I think you mean smaller than 3 kDa, not larger than 3 kDa.

Review form: Reviewer 2**Is the manuscript scientifically sound in its present form?**

No

Are the interpretations and conclusions justified by the results?

No

Is the language acceptable?

Yes

Is it clear how to access all supporting data?

Not Applicable

Do you have any ethical concerns with this paper?

No

Have you any concerns about statistical analyses in this paper?

I do not feel qualified to assess the statistics

Recommendation?

Major revision is needed (please make suggestions in comments)

Comments to the Author(s)

It is suggested that the methods section of the article should be revised to improve its clarity.

Decision letter (RSOS-181433.R0)

21-Nov-2018

Dear Dr Schroeter,

The editors assigned to your paper ("A proteomic method to extract, concentrate, digest, and enrich peptides from fossils with high humic content for mass spectrometry analyses") have now received comments from reviewers. We would like you to revise your paper in accordance with the referee and Associate Editor suggestions which can be found below (not including confidential reports to the Editor). Please note this decision does not guarantee eventual acceptance.

Please submit a copy of your revised paper before 14-Dec-2018. Please note that the revision deadline will expire at 00.00am on this date. If we do not hear from you within this time then it will be assumed that the paper has been withdrawn. In exceptional circumstances, extensions may be possible if agreed with the Editorial Office in advance. We do not allow multiple rounds of revision so we urge you to make every effort to fully address all of the comments at this stage. If deemed necessary by the Editors, your manuscript will be sent back to one or more of the original reviewers for assessment. If the original reviewers are not available, we may invite new reviewers.

- Data accessibility

If you wish to submit your supporting data or code to Dryad (<http://datadryad.org/>), or modify your current submission to dryad, please use the following link:
<http://datadryad.org/submit?journalID=RSOS&manu=RSOS-181433>

- Competing interests

- Authors' contributions

- Acknowledgements

- Funding statement

on behalf of Prof Kevin Padian (Subject Editor)
openscience@royalsociety.org

Comments to Author:

Reviewers' Comments to Author:
Reviewer: 1

Comments to the Author(s)

Schroeter et al. build in the current manuscript on their earlier works in an effort to mitigate a potentially major issue in palaeoproteomics - the presence of large amounts of humics in fossil bone interfering during protein extraction and MS analysis. This is a relevant question to resolve, as is indicated by the recent literature on this topic as well.

The major issue with the manuscript is that an internal control is lacking. That is, claims of relevance made in the paper of enhanced protein extraction through either Method 1 and Method 2 refer to extractions and data presented in a previous paper (Cleland et al., 2016, PRSB; Cleland et al., 2012, Plos One). However, recent research in protein preservation has shown remarkable variation of within-bone protein preservation, necessitating that extraction comparisons are performed on homogenized bone powders. Further issues with comparing the enhanced results derive from an absence in protein or peptide quantification, differences in MS set-up, and column chemistry. Therefore, the main conclusion of the paper could be seen as suggestive.

Minor comments:

-Lines 54-65. This is one massive sentence. Please break up in smaller sentences.

-Lines 85-98. The four elements of the used extraction methods are presented as novel. However, each of these has already appeared in the palaeoproteomic literature at least once. To avoid the suggestion that all four elements are introduced here for the first time, references to relevant literature are justified.

-Line 96: The re-occurring claim that this bone specimen has a particularly high humic content requires quantification.

-Line 126 vs line 147: You mention "a small piece", but later refer to approx. 250 mg., which is a large piece of bone, especially of this age. These two statements are in conflict, especially as "small" frequently refers to samples under 20mg in the isotope, collagen, and bone proteome literature.

-Line 257: I think you mean smaller than 3 kDa, not larger than 3 kDa.

Reviewer: 2

Comments to the Author(s)

It is suggested that the methods section of the article should be revised to improve its clarity.

Author's Response to Decision Letter for (RSOS-181433.R0)

See Appendix A.

RSOS-181433.R1 (Revision)

Review form: Reviewer 2

Is the manuscript scientifically sound in its present form?

No

Are the interpretations and conclusions justified by the results?

No

Is the language acceptable?

Yes

Is it clear how to access all supporting data?

Yes

Do you have any ethical concerns with this paper?

Yes

Have you any concerns about statistical analyses in this paper?

No

Recommendation?

Major revision is needed (please make suggestions in comments)

Comments to the Author(s)

The additional supplemental table does not address the reviewer's previously expressed concerns.

Review form: Reviewer 3

Is the manuscript scientifically sound in its present form?

Yes

Are the interpretations and conclusions justified by the results?

Yes

Is the language acceptable?

Yes

Is it clear how to access all supporting data?

Yes

Do you have any ethical concerns with this paper?

No

Have you any concerns about statistical analyses in this paper?

No

Recommendation?

Accept with minor revision (please list in comments)

Comments to the Author(s)

I have reviewed the revisions made by the authors in order to address the first round of reviews. The authors have responded in an adequate way to all queries.

I do not wish to add any more comments as it would not be fair on the authors, but I would strongly suggest to change the title as follows "A proteomic method to extract, concentrate, digest and enrich peptides from from sub-fossil bones with colored (humic) substances for mass spectrometry analyses". This is because: a) the paper only deals with bone, which has different characteristics than other substrates for paleoproteomics (for example skipping demineralization is not appropriate in all cases); b) "fossil" is a difficult word and often implies no organics and old age - using "sub-fossil" is useful to indicate that you do have organics; c) "abundant" still implies quantitation, which is not available in the paper; d) technically the authors do not provide chemical evidence that the colored substances are in fact "humics" - better to focus on the color aspect.

Decision letter (RSOS-181433.R1)

12-Mar-2019

Dear Dr Schroeter:

Manuscript ID RSOS-181433.R1 entitled "A proteomic method to extract, concentrate, digest, and enrich peptides from fossils with abundant humics for mass spectrometry analyses" which you submitted to Royal Society Open Science, has been reviewed. The comments of the reviewer(s) are included at the bottom of this letter.

Please submit a copy of your revised paper before 04-Apr-2019. Please note that the revision deadline will expire at 00.00am on this date. If we do not hear from you within this time then it will be assumed that the paper has been withdrawn. In exceptional circumstances, extensions may be possible if agreed with the Editorial Office in advance. We do not allow multiple rounds of revision so we urge you to make every effort to fully address all of the comments at this stage. If deemed necessary by the Editors, your manuscript will be sent back to one or more of the original reviewers for assessment. If the original reviewers are not available we may invite new reviewers.

To revise your manuscript, log into <http://mc.manuscriptcentral.com/rsos> and enter your Author Centre, where you will find your manuscript title listed under "Manuscripts with

Decisions." Under "Actions," click on "Create a Revision." Your manuscript number has been appended to denote a revision. Revise your manuscript and upload a new version through your Author Centre.

- Ethics statement

- Data accessibility

- Competing interests

- Authors' contributions

- Acknowledgements

- Funding statement

Kind regards,

Andrew Dunn

on behalf of Prof Kevin Padian (Subject Editor)

Editorial office comments:

Unfortunately, in an earlier round of review, the comments supplied by a reviewer that were intended for the author as well as the Editor were not communicated to the authors, as the comments had been included in the comments to the Editor. Following our standard processes, these comments were not forwarded to the authors (comments supplied in confidence to the Editor are not generally communicated to the authors); however, from the comments of the reviewer, it would seem these comments were intended for the author as well. With this in mind, we now supply the substantive elements of these comments immediately below, with the current round of comments provided below this. We apologise for this state of affairs, and hope you'll address the comments in this round of review.

==Original comments==

there are inconsistencies between different statements in the materials and methods section. For example, on page 6 it is stated that the bone sample was divided into 10 aliquots of which 5 were treated with NaOH and then all 10 samples were treated with the 400/200/4 solution. But on the following page (p7) it is stated that there were three sample sets, one of which had NaOH but no 400/200/4 treatment, which is inconsistent with the description given on the preceding page. Figure 1 is of little help in understanding the experimental protocol. Overall, this article if revised could make a useful contribution to the palaeoproteomics literature but as submitted I think that it fails to meet the journal's requirement that published work should be "scientifically sound, in which the methodology is rigorous and the conclusions fully supported by the data".

Reviewer comments to Author:

Reviewer: 3

Comments to the Author(s)

I have reviewed the revisions made by the authors in order to address the first round of reviews. The authors have responded in an adequate way to all queries.

I do not wish to add any more comments as it would not be fair on the authors, but I would strongly suggest to change the title as follows "A proteomic method to extract, concentrate, digest

and enrich peptides from from sub-fossil bones with colored (humic) substances for mass spectrometry analyses". This is because: a) the paper only deals with bone, which has different characteristics than other substrates for paleoproteomics (for example skipping demineralization is not appropriate in all cases); b) "fossil" is a difficult word and often implies no organics and old age - using "sub-fossil" is useful to indicate that you do have organics; c) "abundant" still implies quantitation, which is not available in the paper; d) technically the authors do not provide chemical evidence that the colored substances are in fact "humics" - better to focus on the color aspect.

Reviewer: 2

Comments to the Author(s)

The additional supplemental table does not address the reviewer's previously expressed concerns.

Author's Response to Decision Letter for (RSOS-181433.R1)

See Appendix B.

RSOS-181433.R2 (Revision)

Review form: Reviewer 4

Is the manuscript scientifically sound in its present form?

Yes

Are the interpretations and conclusions justified by the results?

Yes

Is the language acceptable?

Yes

Do you have any ethical concerns with this paper?

No

Have you any concerns about statistical analyses in this paper?

No

Recommendation?

Accept with minor revision (please list in comments)

Comments to the Author(s)

In the manuscript entitled "A proteomic method to extract, concentrate, digest, and enrich peptides from fossils with colored (humic) substances for mass spectrometry analyses", Schroeter

and co-authors describe an protocol to extract ancient protein residues from bone samples highly infiltrated with coloured (humic) substances. As the authors point out, the challenge the method aims at addressing is significant. The study clearly describes how the method presented clearly leads to higher recoveries in terms of proteins peptides and PSMs. In the current version the manuscript is clearly readable, the experimental protocol is described with enough detail to guarantee its reproducibility and the literature review is pertinent and exhaustive. The evidence presented is solid and reliable and it fully supports the conclusions reported. I have no major comments, just a very few minor suggestions to marginally improve clarity. Specifically:

Line 121: "Two extraction methods were tested; one that..." could be replaced with: "Two extraction methods were tested: one that..."

It could be useful to specify the pH of some of the buffers and solutions used for sample preparation, e.g. ABC.

Line 194: "... an adapted, centrifugal..." could be replaced with: "... an adapted centrifugal...".

Line 317: The title "Efficiency and Modifications" is in my opinion a bit vague and does not effectively describe the content of the following paragraph. I suggest to replace it with something a bit more specific, such as: "Efficiency and Possible Variations of the Experimental Workflow".

Globally, I think the authors did a very good job.

Decision letter (RSOS-181433.R2)

10-Jul-2019

Dear Dr Schroeter:

On behalf of the Editors, I am pleased to inform you that your Manuscript RSOS-181433.R2 entitled "A proteomic method to extract, concentrate, digest, and enrich peptides from fossils with colored (humic) substances for mass spectrometry analyses" has been accepted for publication in Royal Society Open Science subject to minor revision in accordance with the referee suggestions. Please find the referees' comments at the end of this email.

The reviewers and Subject Editor have recommended publication, but also suggest some minor revisions to your manuscript. Therefore, I invite you to respond to the comments and revise your manuscript.

- Ethics statement

- Data accessibility

It is a condition of publication that all supporting data are made available either as supplementary information or preferably in a suitable permanent repository. The data accessibility section should state where the article's supporting data can be accessed. This section should also include details, where possible of where to access other relevant research materials

such as statistical tools, protocols, software etc can be accessed. If the data has been deposited in an external repository this section should list the database, accession number and link to the DOI for all data from the article that has been made publicly available. Data sets that have been deposited in an external repository and have a DOI should also be appropriately cited in the manuscript and included in the reference list.

If you wish to submit your supporting data or code to Dryad (<http://datadryad.org/>), or modify your current submission to dryad, please use the following link:
<http://datadryad.org/submit?journalID=RSOS&manu=RSOS-181433.R2>

- **Competing interests**

- **Authors' contributions**

- **Acknowledgements**

- **Funding statement**

Because the schedule for publication is very tight, it is a condition of publication that you submit the revised version of your manuscript before 19-Jul-2019. Please note that the revision deadline will expire at 00.00am on this date. If you do not think you will be able to meet this date please let me know immediately.

on behalf of Kevin Padian (Subject Editor)
openscience@royalsociety.org

Reviewer comments to Author:
Reviewer: 4

Comments to the Author(s)

In the manuscript entitled "A proteomic method to extract, concentrate, digest, and enrich peptides from fossils with colored (humic) substances for mass spectrometry analyses", Schroeter and co-authors describe an protocol to extract ancient protein residues from bone samples highly infiltrated with coloured (humic) substances. As the authors point out, the challenge the method

aims at addressing is significant. The study clearly describes how the method presented clearly leads to higher recoveries in terms of proteins peptides and PSMs. In the current version the manuscript is clearly readable, the experimental protocol is described with enough detail to guarantee its reproducibility and the literature review is pertinent and exhaustive. The evidence presented is solid and reliable and it fully supports the conclusions reported. I have no major comments, just a very few minor suggestions to marginally improve clarity. Specifically:

Line 121: "Two extraction methods were tested; one that..." could be replaced with: "Two extraction methods were tested: one that..."

It could be useful to specify the pH of some of the buffers and solutions used for sample preparation, e.g. ABC.

Line 194: "... an adapted, centrifugal..." could be replaced with: "... an adapted centrifugal..."

Line 317: The title "Efficiency and Modifications" is in my opinion a bit vague and does not effectively describe the content of the following paragraph. I suggest to replace it with something a bit more specific, such as: "Efficiency and Possible Variations of the Experimental Workflow".

Globally, I think the authors did a very good job.

Author's Response to Decision Letter for (RSOS-181433.R2)

See Appendix C.

Decision letter (RSOS-181433.R3)

23-Jul-2019

Dear Dr Schroeter,

I am pleased to inform you that your manuscript entitled "A proteomic method to extract, concentrate, digest, and enrich peptides from fossils with colored (humic) substances for mass spectrometry analyses" is now accepted for publication in Royal Society Open Science.

Kind regards,

on behalf of Kevin Padian (Subject Editor)
openscience@royalsociety.org

Appendix A

We thank both reviewers for their comments and suggestions. Below, we have detailed the edits we've made to the manuscript in response to their suggestions. Additionally, we have uploaded a "tracked changes" version of the current manuscript that shows all of these changes in the document (as well a few minor typos we have corrected).

Reviewer: 1

Comments to the Author(s)

Schroeter et al. build in the current manuscript on their earlier works in an effort to mitigate a potentially major issue in palaeoproteomics - the presence of large amounts of humics in fossil bone interfering during protein extraction and MS analysis. This is a relevant question to resolve, as is indicated by the recent literature on this topic as well.

The major issue with the manuscript is that an internal control is lacking. That is, claims of relevance made in the paper of enhanced protein extraction through either Method 1 and Method 2 refer to extractions and data presented in a previous paper (Cleland et al., 2016, PRSB; Cleland et al., 2012, Plos One). However, recent research in protein preservation has shown remarkable variation of within-bone protein preservation, necessitating that extraction comparisons are performed on homogenized bone powders. Further issues with comparing the enhanced results derive from an absence in protein or peptide quantification, differences in MS set-up, and column chemistry. Therefore, the main conclusion of the paper could be seen as suggestive.

We thank reviewer 1 for their comments. We agree that a direct comparison is needed to quantify the relative contributions the many differences between the old study and the new study each had to the increase in protein identification, and to account for intra-bone variation. However, the main contribution of this paper is that this combined method allows protein extracts dark with humics to be concentrated and clarified *without* losing low-abundance proteins (e.g., NCPs). Thus, to address the reviewer's concerns we have softened the language to shift focus from relative "improvement" over an older method, to the fact that this combined workflow allows sensitive analyses of low-abundance proteins in fossils containing abundant humics (as supported by its retrieval of 12 non-collagenous proteins). Additionally, we have added a statement acknowledging the limits of the comparison that can be made between the two methods.

Modifications to the text include:

Abstract—We have changed "This method allows better and more sensitive analyses of low-abundance proteins in fossils containing humics" to "This workflow allows analyses of low-abundance proteins in fossils containing humics."

Results—We have added the following sentence to the paragraph comparing the 2015 study with the current study:

"A direct comparison of the HCl-ABC method employed in Cleland 2015¹ and this combined method is needed to quantify the relative contributions to the observed increase in efficiency from the various differences in their workflows (e.g., extraction buffers, MS instrument, column packing) and to fully account for possible intra-bone variation in

protein preservation in this specimen.²⁻³ Regardless, this combined workflow obtained 15 additional proteins previously unreported for this specimen, supporting its overall improvement in recovery.”

Conclusion—We have changed “Because this method allows better and more sensitive MS analyses of low-abundance proteins in fossils with a high humic content” to “Because this workflow allows MS analyses of low-abundance proteins in fossils with abundant humic content”

Minor comments:

-Lines 54-65. This is one massive sentence. Please break up in smaller sentences.

We have separated this list into individual sentences. That section now reads as follows:

“For example, these substances interfere with colorimetric assays used for protein quantitation (e.g., Bradford and bicinchoninic acid (BCA) assays),⁸ preventing accurate measurement of protein for subsequent analyses. They also bind silver-stain, generating false-positives in SDS-PAGE gels, or can pre-stain gel lanes and obscure chemical stains applied to separated proteins.⁹⁻¹⁰ They can contaminate the isotopic content of ancient bone and interfere with stable isotope analyses.^{2, 11} In mass spectrometry experiments, humic substances can cause ion-suppression during electrospray ionization (ESI),^{3, 12} leading to either depressed signal intensity or no signal (regardless of the presence of protein in the sample). Further, they can clog analytical columns or spray tips during liquid chromatography,³ causing unstable spray and/or the loss of samples, columns, or spray tips. Even when these substances do not clog analytical columns, they can generate a build-up of particulate matter on the ion optics, which can suppress or prevent signal during analyses of all subsequent samples, necessitating expensive and time-consuming instrument cleaning.”

-Lines 85-98. The four elements of the used extraction methods are presented as novel. However, each of these has already appeared in the palaeoproteomic literature at least once. To avoid the suggestion that all four elements are introduced here for the first time, references to relevant literature are justified.

To clarify that this is a method combined from elements previously utilized in paleoproteomics and adapted for humic removal, we have added the following sentence to the introduction:

“Although the individual components of this method have been variously applied in previous paleoproteomic studies (e.g., non-demineralizing buffer,⁴⁻⁵ FASP⁶⁻⁷) their combination and adaptation for humic removal has not yet been explored.”

Additionally, because during the course of this review a new paper was published discussing the removal of humics from fossil bones for proteomics, we have added the following sentence to the paragraph of the introduction that talks about previous methods to remove humics from protein extracts:

“Cleland et al.⁵ employed a method utilizing both a non-demineralizing buffer and magnetic beads to separate humics from proteins in unconcentrated protein extracts.”

-Line 96: The re-occurring claim that this bone specimen has a particularly high humic content requires quantification.

A quantitative comparison of the humic content of this moa fossil versus typical fossil specimens is beyond the scope of this paper. Thus, we have modified sentences in the abstract, introduction, and conclusion to avoid suggesting the humic content of this fossil is exceptionally high rather than simply abundant.

Modifications to the text include:

- *Title*—We have changed “fossils with high humic content” to “fossils with abundant humics”
- *Abstract*—We have changed the phrase “a moa fossil with very high humic content” to “a moa fossil dark with humic content”
- *Abstract*—We have changed the phrase “in fossils with high humic content” to “fossils with humic content”
- *Introduction*—We have changed the phrase “a moa fossil with high humic content” to “a moa fossil with humic content”
- *Conclusion*—We have changed the phrase “protein extracts from a moa fossil with a very high humic content” to “protein extracts from a moa fossil inferred from their dark color to have a high humic content”
- *Conclusion*—We have changed “fossils with high humic content” to “fossils with abundant humic content.”

-Line 126 vs line 147: You mention "a small piece", but later refer to approx. 250 mg., which is a large piece of bone, especially of this age. These two statements are in conflict, especially as "small" frequently refers to samples under 20mg in the isotope, collagen, and bone proteome literature.

We have removed “small” from line 126, and now simply refer to “a piece of dark cortical bone.”

-Line 257: I think you mean smaller than 3 kDa, not larger than 3 kDa.

We did indeed mean “less than.” We have corrected this typo.

1. Cleland, T. P.; Schroeter, E. R.; Schweitzer, M. H., Biologically and diagenetically derived peptide modifications in Moa collagens. *Proceedings of the Royal Society B* **2015**, *282* (20150015).
2. Cleland, T. P.; Schroeter, E. R., A comparison of common mass spectrometry approaches for paleoproteomics. *Journal of Proteome Research* **2018**, *17*, 936–945.
3. Simpson, J. P.; Penkman, K. E. H.; Demarchi, B.; Koon, H.; Collins, M. J.; Thomas-Oates, J.; Shapiro, B.; Stark, M.; Wilson, J., The effects of demineralisation and sampling point variability on the measurement of glutamine deamidation in type I collagen extracted from bone. *Journal of Archaeological Science* **2016**, *69*, 29-38.
4. Cleland, T. P.; Schroeter, E. R.; Feranec, R. S.; Vashishth, D., Peptide sequences from the first *Castoroides ohioensis* skull and the utility of old museum collections for palaeoproteomics. *Proceedings of the Royal Society B* **2016**, *283*, 20160593.
5. Cleland, T. P., Human Bone Paleoproteomics Utilizing the Single-Pot, Solid-Phase-Enhanced Sample Preparation Method to Maximize Detected Proteins and Reduce Humics. *Journal of Proteome Research* **2018**, *17* (11), 3976-3983.

6. Cappellini, E.; Gentry, A.; Palkopoulou, E.; Ishida, Y.; Cram, D.; Roos, A.-M.; Watson, M.; Johansson, U. S.; Fernholm, B.; Agnelli, P.; Barbagli, F.; Littlewood, D. T. J.; Kelstrup, C. D.; Olsen, J. V.; Lister, A. M.; Roca, A. L.; Dalén, L.; Gilbert, M.; P., T., Resolution of the type material of the Asian elephant, *Elephas maximus* Linnaeus, 1758 (Proboscidea, Elephantidae). *Zoological Journal of the Linnean Society* **2014**, *170* (1), 222–232.
7. Warinner, C.; Rodrigues, J. F. M.; Vyas, R.; Trachsel, C.; Shved, N.; Grossmann, J.; Radini, A.; Hancock, Y.; Tito, R. Y.; Fiddyment, S.; Speller, C.; Hendy, J.; Charlton, S.; Luder, H. U.; Salazar-García, D. C.; Eppler, E.; Seiler, R.; Hansen, L. H.; Castruita, J. A. S.; Barkow-Oesterreicher, S.; Teoh, K. Y.; Kelstrup, C. D.; Olsen, J. V.; Nanni, P.; Kawai, T.; Willerslev, E.; von Mering, C.; Lewis Jr, C. M.; Collins, M. J.; Gilbert, M. T. P.; Rühli, F.; Cappellini, E., Pathogens and host immunity in the ancient human oral cavity. *Nature Genetics* **2014**, *46* (4), 336–344.

Reviewer: 2

Comments to the Author(s)

It is suggested that the methods section of the article should be revised to improve its clarity. We thank reviewer 2 for this recommendation. Because the reviewer did not provide specific instructions or recommendations, to improve clarity, we constructed a supplemental table (Supplemental Table 1) that breaks down each phase of the extraction and sample preparation procedures into a series of steps. The location of each action (e.g., in tube, in filter), the reagent used and the volume of it added, the incubation time and temperature, the centrifugation time and speed, and the number of times an action is performed/repeated, are listed in a concise format, to complement the more detailed description provided in the text.

To direct readers to this table, we have added the sentence, “Supplementary Table 1 provides a concise breakdown of the steps comprising the extraction, FASP, and stage tipping procedures,” to the end of the first paragraph in the Protein Extraction Section of the Methods.

Additionally, the previous Supplementary Table 1 has now been changed to Supplementary Table 2.

A proteomic method to extract, concentrate, digest, and enrich peptides from fossils with
abundant humics for mass spectrometry analyses

Elena R. Schroeter,^{1*} Kevin Blackburn,² Michael B. Goshe,² Mary H. Schweitzer¹

1. Department of Biological Sciences, North Carolina State University, Raleigh, NC 27513,
easchroe@ncsu.edu

2. Department of Molecular and Structural Biochemistry, North Carolina State University,
Raleigh, NC 27513

E. Schroeter 12/12/2018 2:34 PM
Deleted: high

E. Schroeter 12/12/2018 2:35 PM
Deleted: content

ABSTRACT

Humic substances are break-down products of decaying organic matter that co-extract with
proteins from fossils. These substances are difficult to separate from proteins in solution, and
interfere with analyses of fossil proteomes. We introduce a method combining multiple recent
advances in extraction protocols to both concentrate proteins from fossil specimens with high
humic content, and remove humics, producing clean samples easily analyzed by mass
spectrometry (MS). This method includes: 1) a non-demineralizing extraction buffer that
eliminates protein loss during the demineralization step in routine methods; 2) filter-aided
sample preparation (FASP) of peptides, which concentrates and digests extracts in one filter,
allowing the separation of large humics after digestion; 3) centrifugal stage-tipping, which
further clarifies and concentrates samples in a uniform process performed simultaneously on
multiple samples. We apply this method to a moa fossil (~800–1000 yr) dark with humic
content, generating colorless samples and enabling the detection of more proteins with greater
sequence coverage than previous MS analyses on this same specimen. This workflow allows
analyses of low-abundance proteins in fossils containing humics, and thus may widen the range
of extinct organisms and regions of their proteomes we can explore with MS.

Keywords: paleoproteomics, fossils, humic substances, protein, mass spectrometry, moa

BACKGROUND

E. Schroeter 11/30/2018 1:06 PM

Deleted: very high

E. Schroeter 12/2/2018 2:59 PM

Deleted: method

E. Schroeter 12/2/2018 2:59 PM

Deleted: better and more sensitive

E. Schroeter 11/30/2018 2:04 PM

Deleted: with high

E. Schroeter 11/30/2018 2:04 PM

Deleted: content

[revised manuscript text omitted]

(<http://dx.doi.org/10.5061/dryad.3tv1523>).

All data files can be reviewed at: <https://datadryad.org/review?doi=doi:10.5061/dryad.3tv1523>.

COMPETING INTERESTS

The authors declare they have no competing interests.

AUTHOR'S CONTRIBUTIONS

MHS obtained the specimen, ERS performed molecular lab work and prepared samples, ERS,

KB, and MBG performed MS analyses, ERS analyzed and interpreted data, and drafted the

manuscript. All authors contributed to experimental design and gave final approval for

publication.

ACKNOWLEDGEMENTS

We thank W. Zheng for logistical support and T. Cleland and two anonymous reviewers for

helpful comments on earlier versions of this manuscript.

FUNDING

This work was funded by an Arnold O. Beckman Postdoctoral Fellowship to ERS, an NSF MRI

grant (DBI-1126244) to MBG and RKB, and a NSF INSPIRE grant (EAR-1344198) to MBG

and MHS.

REFERENCES

1. van Klinken, G. J.; Hedges, R. E. M., Experiments on Collagen-Humic Interactions:
Speed of Humic Uptake, and Effects of Diverse Chemical Treatments. *Journal of Archaeological*
*Science* **1995**, *22* (2), 263-270.

2. Szpack, P.; Kripper, K.; Richards, M. P., Effects of sodium hydroxide treatment and
ultrafiltration on the removal of humic contaminants from archaeological bone. *International*
*Journal of Osteoarchaeology* **2017**, 1–8.

3. Qian, C.; Hettich, R. L., Optimized extraction method to remove humic acid interferences
from soil samples prior to microbial proteome measurements. *Journal of Proteome Research*
**2017**, *16* (7), 2537–2546.

4. Sutton, R.; Sposito, G., Molecular structure in soil humic substances: The new view.
*Environmental Science & Technology* **2005**, *39* (23), 9009–9015.

5. Arenella, M.; Giagnoni, L.; Masciando, G.; Ceccanti, B.; Nannipieri, P.; Renella, G.,
Interactions between proteins and humic substances affect protein identifications by mass
spectrometry. *Biology and Fertility of Soils* **2014**, *50* (447–454).

6. Janos, P., Separation methods in the chemistry of humic substances. *Journal of*
*Chromatography A* **2003**, *983*, 1–18.

7. Shin, H.-S.; Monsallier, J. M.; Choppin, G. R., Spectroscopic and chemical
characterizations of molecular size fractionated humic acid. *Talanta* **1999**, *50*, 641–647.

- 8. Redmile-Gordon, M. A.; Armenise, E.; White, R. P.; Hirsch, P. R.; Goulding, K. W. T.,
A comparison of two colorimetric assays, based on Lowry and Bradford techniques, to estimate
total protein in soil extracts. *Soil Biology & Biochemistry* **2013**, *67*, 166–173.
- 9. Dunkelog, R.; Rüttinger, H.-H.; Peisker, K., Comparative study for the separation of
aquatic humic substances by electrophoresis. *Journal of Chromatography A* **1997**, *777*, 355–362.
- 10. Evans, R. D.; Villeneuve, J. Y., A method for characterization of humic and fulvic acids
by gel electrophoresis laser ablation inductively coupled plasma mass spectrometry. *Journal of*
*Analytical Atomic Spectrometry* **1999**, *15*, 157–161.
- 11. van Klinken, G. J.; Bowles, A. D.; Hedges, R. E. M., Radiocarbon dating of peptides
isolated from contaminated fossil bone collagen by collagenase digestion and reversed-phase
chromatography. *Geochimica et Cosmochimica Acta* **1994**, *58* (11), 2543–2551.
- 12. Piccolo, A.; Spiteeller, M., Electrospray ionization mass spectrometry of terrestrial humic
substances and their size fractions. *Analytical and Bioanalytical Chemistry* **2003**, *377*, 1047–
1059.
- 13. Cleland, T. P., Human Bone Paleoproteomics Utilizing the Single-Pot, Solid-Phase-
Enhanced Sample Preparation Method to Maximize Detected Proteins and Reduce Humics.
*Journal of Proteome Research* **2018**, *17* (11), 3976–3983.
- 14. Cleland, T. P.; Vashishth, D., Bone protein extraction without demineralization using
principles from hydroxyapatite chromatography. *Analytical Biochemistry* **2015**, *472*, 62–66.

- 15. Schroeter, E. R.; DeHart, C. J.; Schweitzer, M. H.; Thomas, P. M.; Kelleher, N., Bone
protein "extractomics": Comparing the efficiency of bone protein extractions of *Gallus gallus* in
tandem mass spectrometry, with an eye towards paleoproteomics. *PeerJ* **2016**, *4*, e2603.
- 16. Wisniewski, J. R.; Zougman, A.; Nagaraj, M., Universal sample preparation method for
proteome analyses. *Nature Methods* **2009**, *6* (5), 359–363.
- 17. Yu, Y.; Smith, M.; Pieper, R., A spinnable and automatable StageTip for high throughput
peptide desalting and proteomics. *Protocol Exchange* **2014**, 1–11.
- 18. Cleland, T. P.; Schroeter, E. R.; Feranec, R. S.; Vashishth, D., Peptide sequences from
the first *Castoroides ohioensis* skull and the utility of old museum collections for
palaeoproteomics. *Proceedings of the Royal Society B* **2016**, *283*, 20160593.
- 19. Cappellini, E.; Gentry, A.; Palkopoulou, E.; Ishida, Y.; Cram, D.; Roos, A.-M.; Watson,
487 M.; Johansson, U. S.; Fernholm, B.; Agnelli, P.; Barbagli, F.; Littlewood, D. T. J.; Kelstrup, C.
488 D.; Olsen, J. V.; Lister, A. M.; Roca, A. L.; Dalén, L.; Gilbert, M.; P., T., Resolution of the type
material of the Asian elephant, *Elephas maximus* Linnaeus, 1758 (Proboscidea, Elephantidae).
*Zoological Journal of the Linnean Society* **2014**, *170* (1), 222–232.
- 20. Warinner, C.; Rodrigues, J. F. M.; Vyas, R.; Trachsel, C.; Shved, N.; Grossmann, J.;
Radini, A.; Hancock, Y.; Tito, R. Y.; Fiddyment, S.; Speller, C.; Hendy, J.; Charlton, S.; Luder,
H. U.; Salazar-García, D. C.; Eppler, E.; Seiler, R.; Hansen, L. H.; Castruita, J. A. S.; Barkow-
Oesterreicher, S.; Teoh, K. Y.; Kelstrup, C. D.; Olsen, J. V.; Nanni, P.; Kawai, T.; Willerslev, E.;
von Mering, C.; Lewis Jr, C. M.; Collins, M. J.; Gilbert, M. T. P.; Rühli, F.; Cappellini, E.,

Pathogens and host immunity in the ancient human oral cavity. *Nature Genetics* **2014**, *46* (4),
336–344.

21. Schweitzer, M. H.; Wittmeyer, J. L.; Horner, J. R., Soft tissue and cellular preservation in
vertebrate skeletal elements from the Cretaceous to the present. *Proceedings of the Royal Society*
*B* **2007**, *274*, 183-197.

22. Cleland, T. P.; Schroeter, E. R.; Schweitzer, M. H., Biologically and diagenetically
derived peptide modifications in Moa collagens. *Proceedings of the Royal Society B* **2015**, *282*
(20150015).

23. Cleland, T. P.; Voegelé, K.; Schweitzer, M. H., Empirical Evaluation of Bone Extraction
Protocols. *PLoS ONE* **2012**, *7* (2), e31443.

24. Liu, D.; Liang, L.; Regenstein, J. M.; Zhou, P., Extraction and characterization of pepsin-
solubilized collagen from fins, scales, skins, bones and swim bladders of bighead carp
(*Hypophthalmichthys nobilis*). *Food Chemistry* **2012**, *133*, 1441–1448.

25. Rappsilber, J.; Mann, M.; Ishihama, Y., Protocol for micro-purification, enrichment,
pre-fractionation and storage of peptides for proteomics using StageTips. *Nature Protocols* **2007**,
*2* (8), 1896–1906.

26. Keller, A.; Nesvizhskii, A. I.; Kolker, E.; Aebersold, R., Empirical statistical model to
estimate the accuracy of peptide identifications made by MS/MS and database search. *Analytical*
*Chemistry* **2002**, *74* (20), 5383–5392.

27. Nesvizhskii, A. I.; Keller, A.; Kolker, E.; Aebersold, R., A statistical model for
identifying proteins by tandem mass spectrometry. *Analytical Chemistry* **2003**, *75* (17), 4646–
4658.

28. Liu, H.; Sadygov, R. G.; Yates, J. R. I., A model for random sampling and estimation of
relative protein abundance in shotgun proteomics. *Analytical Chemistry* **2004**, *76*, 4193–4201.

29. Cleland, T. P.; Schroeter, E. R., A comparison of common mass spectrometry approaches
for paleoproteomics. *Journal of Proteome Research* **2018**, *17*, 936–945.

30. Simpson, J. P.; Penkman, K. E. H.; Demarchi, B.; Koon, H.; Collins, M. J.; Thomas-
Oates, J.; Shapiro, B.; Stark, M.; Wilson, J., The effects of demineralisation and sampling point
variability on the measurement of glutamine deamidation in type I collagen extracted from bone.
*Journal of Archaeological Science* **2016**, *69*, 29-38.

31. Lipecka, J.; Chhuon, C.; Bourderioux, M.; Bessard, M.-A.; van Endert, P.; Edelman, A.;
Guerrero, I. C., Sensitivity of mass spectrometry analysis depends on the shape of the filtration
unit used for filter aided sample preparation (FASP). *Proteomics* **2016**, *16*, 1852–1857.

Figure 1. Diagram illustrating the extraction workflow of Method 2 (400/200/4 only). NaOH pre-
treatment step of Method 1 is not illustrated; however, NaOH supernatants were also
concentrated, digested, and stage tipped as shown.

Figure 2. Progression of extracts through preparation for MS. (Row 1) Supernatant (NaOH or
400/200/4) immediately upon collection. Each tube represents one of five that were subsequently
concentrated together. (Row 2) Supernatants after concentration and FASP using 3 kDa MWCO

filters. Despite combining five tubes with high humic content, passing samples through the filter
after digestion removed a large portion of the high-weight humics from the smaller peptides.
(Row 3) Supernatants after stage tipping. Upon elution from the stage tips, samples were nearly
colorless, suggesting that most interfering humic substances have been removed.

Figure 3. Concentrated 400/200/4 extract, before FASP digestion. Extracts dark with humic
substances became increasingly dark and opaque during concentration, indicating that humic
substances were too large to pass through the filter and co-concentrated with proteins. After
proteolytic digestion, these humics remained in the filter while digested peptides passed through.

Figure 4. Comparison of the number of unique peptides (A) and sequence coverage (B)
identified for each protein between Method 1 (NaOH + NaOH-400/200/4 fractions) and Method
2 (400/200/4 only). Both methods recover nearly identical amounts of unique peptides and
sequence coverage for most proteins, suggesting that the inclusion of a NaOH pretreatment step
in Method 1 does not provide additional proteome information over the sole 400/200/4
incubation of Method 2. These data represent the total, non-overlapping peptides and coverage
relative to the mature form of the protein (i.e., without pro-peptide regions), combined from all
three injections of each sample. See STable 1 for more details.

Appendix B

We thank both reviewers for their comments and suggestions. Below, we have detailed the edits we've made to the manuscript.

Reviewer: 2

There are inconsistencies between different statements in the materials and methods section. For example, on page 6 it is stated that the bone sample was divided into 10 aliquots of which 5 were treated with NaOH and then all 10 samples were treated with the 400/200/4 solution. But on the following page (p7) it is stated that there were three sample sets, one of which had NaOH but no 400/200/4 treatment, which is inconsistent with the description given on the preceding page. Figure 1 is of little help in understanding the experimental protocol. Overall, this article if revised could make a useful contribution to the palaeoproteomics literature but as submitted I think that it fails to meet the journal's requirement that published work should be "scientifically sound, in which the methodology is rigorous and the conclusions fully supported by the data."

We apologize for the lack of clarity on how our sample sets were obtained. First, we will explain here where we think the confusion arose from:

This paper tests two different methods for extraction.

- **Method 1:** incubates 5 aliquots of bone in NaOH, then collects and pools that NaOH. Then, the 400/200/4 solution is added to the same 5 aliquots of NaOH-treated bone, which is then collected and pooled into a separate sample. This generates 2 types of samples; a NaOH sample that was incubated with fresh, untreated bone (because it was the first step), and a 400/200/4 sample that is derived from the same, now NaOH-treated bone.
- **Method 2:** incubates 5 aliquots of fresh, untreated bone in the 400/200/4 solution and collects it. This generates 1 type of sample, a 400/200/4 sample that is derived from fresh, untreated bone.

Overall, Method 1 generates two sample types (NaOH treatment and NaOH-400/200/4 treatment) and Method 2 only one sample type (400/200/4 treatment). This is how we generated 3 sample types from 2 methods, and this is how we have a sample that has NaOH with no 400/200/4—it is generated as part of a two step method in which the first step (NaOH) and the second (400/200/4) are analyzed separately. Importantly, the NaOH-400/200/4 sample generated by this method *does not* combine the NaOH fraction and the 400/200/4 fraction of Method 1, it is only labeled this way to distinguish the 400/200/4 sample from bone that has had a NaOH pre-treatment from the 400/200/4 sample from bone that did not have the NaOH pre-treatment (Method 2).

We agree this has the potential to be confusing, and is not elucidated by the workflow figure we provided previously. Thus, to illustrate the origin of these three samples more plainly, we have added the following diagram and figure caption to the supplemental material:

Figure S1. Diagram of proteomic samples generated by this study. (A) Method 1: 5 aliquots of 50 mg ground moa bone powder were incubated in 0.1 M NaOH. The NaOH supernatant was subsequently collected and concentrated into a single sample, “NaOH.” These NaOH-treated bone pellets were then incubated in 400/200/4 solution, which was collected and concentrated into a second sample, “NaOH-400/200/4.” This label denotes that it was generated from bone that received 400/200/4 *after* pretreatment with NaOH. (B) Method 2: 5 aliquots of 50 mg ground moa bone powder were incubated directly in 400/200/4. The 400/200/4 supernatant was subsequently collected and concentrated into a solitary sample, “400/200/4.”

To direct readers to this diagram, we have added the following sentence to the first paragraph of the “Protein Extraction” section (lines 130-131):

“Supplementary Figure 1 illustrates a diagram of supernatant collection steps and sample generation for each method.”

We hope that this figure, combined with the previously provided workflow diagram and supplementary table detailing the steps of the procedure, is sufficiently clear for all readers.

Reviewer 3

I have reviewed the revisions made by the authors in order to address the first round of reviews. The authors have responded in an adequate way to all queries.

I do not wish to add any more comments as it would not be fair on the authors, but I would strongly suggest to change the title as follows "A proteomic method to extract, concentrate, digest and enrich peptides from from sub-fossil bones with colored (humic) substances for mass spectrometry analyses". This is because: a) the paper only deals with bone, which has different characteristics than other substrates for paleoproteomics (for example skipping demineralization is not appropriate in all cases); b) "fossil" is a difficult word and often implies no organics and old age - using "sub-fossil" is useful to indicate that you do have organics; c) "abundant" still implies quantitation, which is not available in the paper; d) technically the authors do not provide chemical evidence that the colored substances are in fact "humics" - better to focus on the color aspect.

We thank the reviewer for this suggestion, and have adopted most of these comments into the new title:

“A proteomic method to extract, concentrate, digest, and enrich peptides from fossils with colored (humic) substances for mass spectrometry analyses”

We agree with the reviewer that “fossil” is a difficult word. There is not really a consensus in the paleo community whether “fossil” implies the complete lack of organics, or how much time can be encompassed by the term “sub-fossil.” Thus, we are concerned that if we use the term “sub-fossil” it will give readers the mistaken impression that this method is only applicable to *very* recently extinct species (i.e., the moa itself, as this sample is only 1000 years old). Because proteomic methods are increasingly applied to specimens tens to hundreds of thousands of years old without controversy, which few would categorize as “sub-fossil,” we think it is more appropriate to leave the term “fossil” in the title. We have made the other changes in both the main text and the supplement.

Appendix C

We thank the reviewer for their comments and suggestions. Below, we have detailed the edits we've made to the manuscript in accordance with their recommendations.

Reviewer comments to Author:

Reviewer: 4

Comments to the Author(s)

In the manuscript entitled "A proteomic method to extract, concentrate, digest, and enrich peptides from fossils with colored (humic) substances for mass spectrometry analyses", Schroeter and co-authors describe an protocol to extract ancient protein residues from bone samples highly infiltrated with coloured (humic) substances. As the authors point out, the challenge the method aims at addressing is significant. The study clearly describes how the method presented clearly leads to higher recoveries in terms of proteins peptides and PSMs. In the current version the manuscript is clearly readable, the experimental protocol is described with enough detail to guarantee its reproducibility and the literature review is pertinent and exhaustive. The evidence presented is solid and reliable and it fully supports the conclusions reported. I have no major comments, just a very few minor suggestions to marginally improve clarity.

Line 121: "Two extraction methods were tested; one that..." could be replaced with: "Two extraction methods were tested: one that..."

The semicolon in line 121 has been replaced with a colon.

It could be useful to specify the pH of some of the buffers and solutions used for sample preparation, e.g. ABC.

We agree, and have added the following pH information to the methods section:

- The pH for the 0.1 M NaOH solution has been added to line 127. The sentence now reads: Five tubes of each set received 1 ml of 0.1 M NaOH (pH ~12.8), were vortexed to mix, then were incubated at 4°C for 4 h with rocking.
- The pH for the 400/200/4 solution has been added to line 144. The sentence now reads: "All 10 aliquots of bone powder then received 600 µL of a solution¹⁴ containing 400 mM ammonium phosphate diabolic (APD), 200 mM ammonium bicarbonate (ABC), and 4 M guanidine HCl (GuHCl) (pH 8.2), hereafter referred to as "400/200/4" for brevity."
- The pH for the 50 mM ABC solution has been added to line 164. The sentence now reads: After the last centrifugation, filters containing concentrated sample received 300 µL of 50 mM ABC (pH 7.8) and were then stored overnight at 4°C, in a humidity chamber wrapped in parafilm to prevent filters from drying.

Line 194: "... an adapted, centrifugal..." could be replaced with: "... an adapted centrifugal...".

The comma between 'adapted' and 'centrifugal' in line 194 has been removed.

Line 317: The title "Efficiency and Modifications" is in my opinion a bit vague and does not effectively describe the content of the following paragraph. I suggest to replace it with something a bit more specific, such as: "Efficiency and Possible Variations of the Experimental Workflow".

We agree, and have changed this heading (line 317) to the following:

“Efficiency and Potential Variations of the Experimental Workflow”

Globally, I think the authors did a very good job.

We thank the reviewer for their time and insights!

Editorial Comments:

- Ethics statement

We have added the following section heading and text to the end of the manuscript:

ETHICS

No humans, extant animals, or specimens requiring ethical approval were used in this study.

Thank you for resubmitting your paper. Before we proceed, we require for you to include a reference to your Dryad dataset in the reference list of your manuscript. Please include this reference, along with the DOI and both the 'for review' and 'for publication' URLs where they differ.

We have added a citation to the Dryad dataset to line 235 of the manuscript (“Raw data files²⁶ from the LC-MS/MS acquisitions...”). The data set now appears as citation #26 in the reference list.